# Mitochondrial Contribution to Inflammation in Diabetic Kidney Disease

**DOI:** 10.3390/cells11223635

**Published:** 2022-11-16

**Authors:** Alla Mitrofanova, Antonio M. Fontanella, George W. Burke, Sandra Merscher, Alessia Fornoni

**Affiliations:** 1Katz Family Division of Nephrology and Hypertension, Department of Medicine, University of Miami Miller School of Medicine, Miami, FL 33136, USA; 2Peggy and Harold Katz Family Drug Discovery Center, University of Miami Miller School of Medicine, Miami, FL 33136, USA; 3Department of Surgery, University of Miami Miller School of Medicine, Miami, FL 33136, USA

**Keywords:** DKD, inflammation, mitochondria, innate immunity, podocyte, tubule cells

## Abstract

Diabetes is the leading cause of chronic kidney disease worldwide. Despite the burden, the factors contributing to the development and progression of diabetic kidney disease (DKD) remain to be fully elucidated. In recent years, increasing evidence suggests that mitochondrial dysfunction is a pathological mediator in DKD as the kidney is a highly metabolic organ rich in mitochondria. Furthermore, low grade chronic inflammation also contributes to the progression of DKD, and several inflammatory biomarkers have been reported as prognostic markers to risk-stratify patients for disease progression and all-cause mortality. Interestingly, the term “sterile inflammation” appears to be used in the context of DKD describing the development of intracellular inflammation in the absence of bacterial or viral pathogens. Therefore, a link between mitochondrial dysfunction and inflammation in DKD exists and is a hot topic in both basic research and clinical investigations. This review summarizes how mitochondria contribute to sterile inflammation in renal cells in DKD.

## 1. Introduction

Chronic kidney disease (CKD) is a worldwide health issue with an estimated prevalence of 16% [1]. CKD is characterized by proteinuria, reduced glomerular filtration rate and progressive glomerular, tubular and interstitial damage. Glomeruli play an important role in filtering blood, and glomerular injury leads to the development of various glomerular diseases, including diabetic kidney disease (DKD) which is the most common single cause of end stage kidney disease in the United States [2]. Currently, the pharmacological management of DKD includes the use of angiotensin converting enzyme inhibitors or angiotensin receptor blockers in association with sodium–glucose cotransporter 2 inhibitors or non-steroidal mineralocorticoid receptor antagonists [3]. However, these interventions only partially stabilize kidney function [4], and further work is needed to elucidate the precise pathological mechanisms contributing to DKD development and progression.

Among other factors that contribute to the development and progression of glomerular diseases, inflammation, oxidative stress, and immune system activation have recently gained more attention. Inflammation is generally initiated by the activation of pattern recognition receptors (PRRs) that are expressed by immune and non-immune cells [5]. Besides infection-associated molecules, PRRs may be activated by endogenous molecules called damage-associated molecular patterns (DAMPs), which include nucleic acids, ATP and proteins. However, DAMPs contribute to the initiation of inflammatory responses in a state of cellular stress or death when perturbations in permeability of various cellular compartments occur. Therefore, mitochondria, as the evolutionary remnants of ancestral alphaproteobacteria [6], have an important role in controlling cellular inflammation. Moreover, mitochondria are complex organelles and play a significant role in regulating cell death by apoptosis or necrosis [7]. All these facts indicate mitochondria are a unique platform for the redistribution of DAMPs and the activation of PRRs.

Considering that the kidney and heart [8] are the two organs that require the highest mitochondrial content to enable proper function, the role of mitochondria in the development and progression of kidney diseases, including DKD, has become the focus of recent studies [9,10,11]. Podocytes are terminally differentiated epithelial cells that play a key role in the glomerular filtration and are target cells in diabetes-associated kidney injury. Podocytes are highly dynamic and require substantial amounts of energy to maintain proper organization of cytoskeletal and extracellular matrix proteins and for foot processes remodeling [12]. Interestingly, under normal physiological conditions, tubular cells and podocytes use different substrates to produce energy: tubular cells rely on fatty acid β-oxidation, while podocytes preferentially use glucose [13,14,15,16]. However, under disease conditions, such as DKD, podocytes switch their energy substrate to fatty acids, in association with a reduced expression of many glycolytic enzymes, while the expression of β-oxidation enzymes is up-regulated [17]. Although podocytes contain a relatively small number of mitochondria, their contribution to podocyte injury in the state of a cellular stress becomes more obvious, as abnormalities in mitophagy, a mechanism of mitochondrial specific autophagy, and caspase activation can be observed. In this review, we discuss the mechanisms through which mitochondria control the intracellular inflammatory responses in podocytes and we will discuss potential areas that need future investigation.

## 2. Mitochondrial Regulation of Inflammation

### 2.1. cGAS-STING Signaling

Cyclic GMP-AMP (cGAS) is a nuclear and cytosolic protein that senses the presence of double-stranded DNA in the cytosol leading to the formation of a second messenger, cyclic GMP-AMP (cGAMP) and activation of stimulator of interferon genes (STING). STING activation culminates in the recruitment of different kinases: TANK-binding kinase 1 (TBK1), mitogen-activated protein kinase kinase kinase 14 (MAP3K14 or NIK) and heterotrimeric IκB kinase (IKK), which, in turn, promote interferon regulatory factor 3 (IRF3), non-canonical nuclear factor kappa B (NF-κB) and canonical NF-κB, respectively. As a result, IRF3 activation results in type I interferon responses, which are usually mainly associated with antiviral and anticancer effects [18], while NF-κB activation may lead to a broad spectrum of effects [19,20,21]. A common concept suggests that STING predominantly localizes to the outer membrane of the endoplasmic reticulum, but some studies report the presence of STING in the mitochondrial membrane [22,23].

While the cGAS-STING pathway was originally discovered in the context of the innate immunity response to infections and cancer [24,25,26], it is clear that the STING pathway is more than just important in pathogen detection; it also plays a significant role in the detection of self-DNA released from damaged mitochondria (Figure 1), dying cells or tumor cells. Thus, activation of the cGAS-STING in response to the presence of mitochondrial DNA (mtDNA) present in the cytosol of cells has been shown in mouse models of renal fibrosis [27]. Our studies suggest that STING phosphorylation is increased in the db/db mouse model of DKD at baseline and that pharmacological STING inhibition protects from DKD progression [28,29]. Interestingly, induction of STING itself in wildtype mice resulted in proteinuria and podocyte foot process effacement in studies [28,29]. Using eNOS db/db mice and rats with type 2 diabetic nephropathy, others demonstrated increased activity of the cGAS-STING pathway [30]. In patients with DKD, the presence of plasma and urinary mtDNA was recently recognized as a potential marker of early DKD progression [31,32,33,34]. However, the mechanisms of mtDNA escape into the cytosol remain elusive.

mtDNA is particularly vulnerable to damage. Given the fact that mtDNA resides in close proximity to mitochondrial reactive oxygen species (mtROS), oxidative lesions often cause mtDNA damage via DNA strand breakage or nucleotide base oxidation. In DKD, chronic hyperglycemia disrupts the bioenergetic balance, which results in mtROS production and oxidative mtDNA damage as shown in studies of glomeruli from DBA/2J mice [35] and streptozotocin (STZ)-induced DKD rats [36]. Additionally, in STZ-induced DKD mice mitochondrial biogenesis is reduced due to decreased mRNA expression of peroxisome proliferator-activated receptor-gamma coactivator (PGC-1α), nuclear respiratory factor 1 (NRF-1) and mitochondrial transcriptional factor A (TFAM) [37]. Interestingly, decreased expression of TFAM has been reported in renal tissues from patients with CKD stage 4, while TFAM knockdown in mice results in mtDNA leakage into the cytosol and activation of the cGAS-STING pathway [27]. Release of mtDNA into the cytosol can cause activation of the proapoptotic pore-forming proteins BCL2-associated X, apoptosis regulator (BAX) and BCL2 antagonist/killer 1(BAK1) as discussed below. A link between lipotoxicity and mtDNA release into the cytosol followed by activation of the cGAS-STING pathway was recently reported in db/db mice on a high fat diet [38].

In summary, an abundant literature shows the ability of mtDNA to activate the cGAS-STING signaling pathway in the kidney, thereby driving an inflammatory response in DKD.

### 2.2. RLR Signaling

The RIG-I like receptor (RLR) signaling pathway is another inflammatory pathway that can be stimulated by mitochondria (Figure 1). Unlike the cGAS-STING pathway, the RLR pathway is activated by foreign, altered, or ectopic RNA [39]. It has been shown that mitochondrial RNA (mtRNA) can be released into cytosol in a state of reduced polyribonucleotide nucleotidyltransferase 1 (PNPT1) expression, leading to mtRNA degradation, and activation of RLR melanoma differentiation-associated protein 5 (MDA5) [40]. Interestingly, mtDNA double-strand breaks have been shown to contribute to BAX-BAK1-mediated mtRNA release into cytosol with further activation of RIG-I, but not MDA5 [41]. However, the mechanisms leading to the differential activation of RIG-I versus MDA5 by mtRNA species remain to be discovered.

Notably, a recent study of patients with type 2 diabetes revealed a specific serum RNA signature in patients with DKD when compared to patients with diabetes and no DKD, where downregulation of four mitochondrial messenger RNAs (ATP6, ATP8, COX3 and ND1) was found to correlate with serum creatinine and estimated glomerular filtration rate [42]. A genome-wide association study analysis of patients with type 1 diabetes (n = 19,406) also revealed the association of RIG-I/MDA5 and interferon alpha beta gene set [43].

### 2.3. Inflammasome Signaling

The inflammasome comprises a class of signalosomes in innate immunity that promote inflammation and induce an inflammatory form of programmed cell death, called pyroptosis. Early studies have shown that cytosolic mtDNA can also drive the activation of inflammasomes [44], specifically of the inflammasome that contains nucleotide-binding domain-like receptor (NLR) family pyrin domain-containing 3 (NLRP3) as a sensing component (Figure 1). NLRP3 assembles the inflammasome through oligomerization with apoptosis-associated speck-like protein (ASC) to elicit robust caspase 1 (CASP1) activation and production of interleukins 1β (IL-1β) and 18 (IL-18). Oxidized mtDNA release into the cytosol upon mitochondrial dysfunction has been shown to activate the NLRP3 inflammasome [45], and a feedforward loop was identified in which inflammasome activation facilitates mtDNA release via mtROS production [46]. Besides the mtROS-associated NLRP3-ASC-driven mtDNA escape into the cytosol, a physical interaction between NLRP3 and thioredoxin-interacting protein (TXNIP), a nuclear protein that controls the cellular redox state, may induce mitochondrial damage and mtDNA leakage [47]. However, ROS inhibitors seem to disrupt inflammasome priming, i.e., the synthesis of inflammasome components, but not its activation [48]. In line with this notion, recent studies showed that NLRP3 inflammasome activation depends on the mitochondrial electron transport chain [49], and that cytosolic oxidized mtDNA serves as the ultimate NLRP3 ligand [50]. Of note, mitochondrial damage per se does not trigger NLRP3 signaling if priming is omitted (reviewed in [51]). Moreover, NLRP3 inflammasome activation fails under conditions of TFAM deficiency [50]. However, mtDNA has been shown to also activate inflammasomes that use the absence of melanoma 2 (AIM2) as a sensing component [52]. Interestingly, while oxidized DNA seems to activate NLRP3 inflammasomes [45], AIM2-containing counterparts are suggested to recognize non-oxidized DNA [50].

Persistent and aberrant NLRP3 signaling underlies many chronic diseases, including type 1 and type 2 diabetes, while NLRP3 deficiency has been shown to protect against injury, irrespective of the renal cell type [53,54]. mtDNA seems to be one of the main triggers of NLRP3 activation in streptozotocin-induced diabetic [55,56] and in high fat diet mice [57]. Using primary renal tubular epithelial cells and unilateral ureteral obstruction (UUO) mice, a recent study demonstrated that peroxisomal proliferator-γ coactivator-1α (PGC-1α) ameliorates NLRP3 inflammasome-associated renal fibrosis via the modulation of mitochondrial dynamics [58]. NLRP3 activation also contributes to DKD progression as shown in a study demonstrating that podocyte-specific Nlrp3 or caspase-1 deficiency resulted in protection from DKD [59]. In contrast, another group reported that using an NLRP3-specific inhibitor, MCC950, did not confer renoprotective effects using streptozotocin-induced diabetic mice as it did not reduce renal inflammation (glomerular accumulation of CD68 positive cells), mesangial expansion and glomerulosclerosis [60]. However, an earlier study reported that MCC950 lowered fibrosis, renal inflammation and provided protection from kidney failure in a model of oxalate nephropathy [61]. Less is known about the role of AIM2 inflammasomes in diabetes and DKD development. It has been shown that AIM2 inflammasomes directly interact with apoptosis-associated speck-like protein and contribute to the development of many human diseases, including type 2 diabetes, where cell-free mtDNA has been shown to activate AIM2 inflammasomes [62].

In summary, mtDNA is a major DAMP for inflammasome activation that contributes to chronic kidney disease development and progression. Moreover, NLRP3 and AIM2 may represent a potential therapeutic target to ameliorate DKD-associated podocyte and tubular injury.

### 2.4. TLR Signaling

Toll-like receptor (TLR) signaling plays a key role in the innate immune system by recognizing pathogen-associated molecular patterns (PAMPs) leading to the activation of NF-κB and interferon production. The TLR family comprises 10 members in humans (TLR1-TLR10) and 12 members in mice (TLR1-TLR12). TLRs are located on the cell plasma membrane, except for TLR3, TLR7, TLR8 and TRL9, which are found in intracellular vesicles where they sense the nucleic acids inside a cell. Early studies showed that naked as well as protein-bound mtDNA has been shown to activate TLR9 (Figure 1) and advanced glycosylation end product-specific receptor (RAGE) [63,64]. Other studies demonstrated that treatment in vitro or in vivo with mtDNA results in increased levels of TLR9, NF-κB and nuclear factor of kappa light polypeptide gene enhancer in B-cells inhibitor alpha (IκB-α) in different tissues [65,66,67]. De novo TLR9 expression has been shown in podocytes of some patients with glomerular diseases [68,69,70,71], suggesting that endogenous mtDNA serves as a ligand and may facilitate podocyte apoptosis [72]. In a model of acute kidney injury, absence of TLR9 reduced mtDNA-mediated kidney injury [73]. In DKD, the expression of TLR2, TLR4, TLR5, TLR7, TLR8 and TLR9 has been described, but TLR2 and TLR4 are the two most extensively studied receptors (reviewed in [74]). Sparse studies on TLR3 and TLR9 in DKD suggest that in the ApoE-/- streptozotocin-induced mouse model of DKD, TLR3 and TLR9 are activated in the kidney [75]. Similarly, enhanced expression of TLR3 was reported in tubules from patients with DKD [76]. Nevertheless, no data are available that would connect activation of TLR3 and TLR9 in DKD with the release of mtDNA into the cytosol, which may be the subject of future studies.

### 2.5. NF-κB Signaling

Nuclear factor-κB (NF-κB) represents a family of transcription factors which consists of five structurally related members (NF-κB1, NF-κB2, RelA, RelB and c-Rel) and regulates a large array of genes involved in the regulation of the immune and inflammatory responses. The activation of the NF-κB involves two major signaling pathways: (1) in the canonical pathway, NF-κB responds to diverse stimuli, including cytokine receptors, PRRs, TNF receptors, T-cell and B-cell receptors [77]; (2) in the noncanonical (alternative) pathway, NF-κB selectively responds to specific ligands such as lymphotoxin beta receptor (LTβR), tumor necrosis factor receptor superfamily member 13C (TNFRSF13C or BAFFR), CD40, or RANK [78]. Functionally, the canonical NF-κB pathway is involved in almost all aspects of the immune response, while the noncanonical NF-κB appears to be involved in the regulation of specific functions of the adaptive immune system. Interestingly, the suppression of inhibitor of apoptosis (IAP) proteins by the cytosolic mitochondrial protein SMAC shifts NF-κB signaling from the canonical to the noncanonical pathway upon stabilization of mitogen-activated protein kinase kinase kinase 14 (MAP3K14), and this process is orchestrated by BAK1-BAX oligomers [79].

A recent study addressing the role of NF-κB in DKD showed an activation of the NF-κB pathway in diabetic rats that progress to DKD, whereby inflammation was restricted to the glomerular compartment with intense glomerular macrophage infiltration [80]. Earlier studies also confirmed modest activation of the glomerular NF-κB signaling pathway in streptozotocin treated rats as early as one month after the induction of diabetes [81], while in patients with type 2 diabetes, NF-κB activation was mainly detected in cortical tubular epithelial cells and, to a lesser extent, in some glomeruli [82]. Similarly, in patients with type 1 diabetes and DKD, p65 positive glomeruli and inflammation in the area of the renal interstitium were found [80]. High glucose has also been shown to induce NF-κB activation and upregulation of proinflammatory cytokines in human proximal tubular epithelial cells [83] and in podocytes [84]. Interestingly, long-term (12 months) NF-κB inhibition in diabetic rats using pyrrolidine dithiocarbamate resulted in reduced IL-6 production and prevented the development of glomerulosclerosis and loss of podocyte integrity in one study [80]. Therefore, the canonical pathway seems to be the prevalent NF-κB activation pathway in DKD (Figure 2). However, the exact mechanisms leading to NF-κB activation in DKD remain unclear and require further investigation.

To make the picture even more complicated, NF-κB subunits (IkBα, p65) and NF-κB pathway proteins (IKKα, IKKβ and IKKγ) are present in the inner mitochondria matrix [85,86,87,88]. Collectively, these studies suggest that NF-κB can non-specifically bind mtDNA sequences and regulate mRNA expression of a variety of target genes. Recent studies also demonstrated that NF-κB is involved in mitochondrial fission [89], regulation of BAX mediated cytochrome c release to control apoptosis [90], organization of the energy metabolism network by controlling the balance between glycolytic utilization and mitochondrial respiration [42,91], and in controlling respiratory chain gene expression including the expression of COXI, COXIII and CytB [86,92,93]. Moreover, NF-κB p62 induction restricts NLRP3 inflammasome activation via the elimination of damaged mitochondria [94].

## 3. Mitochondrial Regulation of Cell Death

### 3.1. BAX-BAK1 Signaling

One of the mechanisms of mtDNA release in the course of mitochondrial outer membrane permeabilization involves the proapoptotic pore-forming proteins BCL2-associated X, apoptosis regulator (BAX) and BCL2 antagonist/killer 1 (BAK1). Under physiological conditions, mitochondrial outer membrane permeabilization is actively prevented by anti-apoptotic molecules BCL2, BCL2-like protein 1 (BCL2-L1, best known as BCL-XL) and MCL1. In the presence of an apoptotic stimulus such as BH3-interacting domain death agonist (BID) or BCL2-binding component 3 (BBC3, best known as PUMA), displacement of BAX and BAK1 from inhibitory interactions with BCL2, BCL-XL or MCL1 occurs. This, in turn, results in the translocation of cytochrome c from the mitochondrial intermembrane space into cytosol, assembly of an apoptotic peptidase-activating factor 1 (APAF1) and caspase 9 (CASP9)-containing molecular complex (known as apoptosome) and activation of CASP3 as one of the final steps in the apoptotic cascade. Intriguingly, BAX- and BAK1-independent mtDNA release have recently been described. Thus, proteolytically activated BID has been shown to form pores in mitochondria, independently of BAX and BAK1 in human cells [95]. In another study, mild mitochondrial stress did not result in mitochondrial outer membrane permeabilization and mtDNA release was associated with a voltage-dependent anion channel (VDAC)-dependent mechanism [96,97]. Even more interesting, different isoforms of VDAC have been shown to be associated with mitochondrial permeability transition (mPT) (reviewed in [98]), a regulated process of mitochondrial matrix swelling leading to abrupt loss of the impermeability of the inner mitochondrial membrane. This transition is mediated by the mitochondrial permeability transition pore (mPTP), a mitochondrial protein complex, by changing conformation and forming an IMM pore in response to some stimuli. The specific protein components and exact mechanisms of pore formation are poorly understood, but inhibition of mPTP opening by TRAP1 has been shown to protect against diabetic renal injury in STZ treated rats [99]. While VDAC isoforms are no longer considered to form part of the mPTP complex itself, its influence on mPT suggests a possible role as an mtDNA pore during mitochondrial swelling.

In the kidney, BAK1 knockout in a human podocyte cell line has been shown to diminish apolipoprotein L1 (APOL1) expression [100], a protein associated with CKD in populations with recent African ancestry. In another study, double knockout of BAX and BAK1 in proximal tubules resulted in decreased apoptosis in renal tubular cells and suppressed renal interstitial fibrosis in a model of unilateral urethral obstruction (UUO) [101]. Using a mouse model of acute kidney injury, BAX and BAK1 knockout was shown to attenuate renal tubular cell apoptosis and decrease cytochrome c release [102]. In support, high glucose-associated activation of BAX results in increased apoptosis of β-cells in mice on a high fat diet, while ablation of the Bax gene in islets improves diabetes [103]. Treatment of podocytes with high glucose (30 mM) also activates BAK1, BAX and cytochrome c, resulting in increased apoptosis [104]. Further, studies in podocytes isolated from a streptozotocin-induced mouse models of DKD and in the mouse podocyte clone 5 (MCP5) cell line revealed the anti-apoptotic protein BCL-2 as an important hub in the regulation of autophagy and apoptosis levels in the presence of high glucose [105]. In the same study decreased levels of BCL-2 were described in patients with diabetic nephropathy.

In summary, increasing studies support the idea that suppression of anti-apoptotic BCL-2 and activation of pro-apoptotic BAK1 and BAX are at the core of a complex intrinsic apoptotic pathway that contributes to DKD progression.

### 3.2. Cardiolipin

The mitochondrial unique phospholipid cardiolipin (CL), which is predominantly found in the inner mitochondrial membrane (IMM) [106], is crucial for biophysical properties of mitochondrial membranes, where it modulates energy production and participates in inflammation, mitophagy and apoptosis [107,108,109,110,111]. Intriguingly, the structural disruptions that accompany late stage regulated cell death generate mitochondrial fragments containing CL in the extracellular microenvironment, where CL promotes increased expression of MHC class I-like molecule CD1d on antigen presenting cells and results in the activation of a cardiolipin-specific population of T cells [112]. Thus, cardiolipin can promote inflammatory responses. In yeast cells, a 30% increase in the CL content results in physical modification of mitochondrial membranes affecting mtDNA stability via physical interaction between CL and mtDNA [113]. In liver, CL-mediated membrane remodeling also results in mtDNA aggregation and release [114].

In DKD, we reported increased CL peroxidation in db/db and ob/ob mice in association with mitochondrial dysfunction, while the inhibition of CL peroxidation with Elamipretide, which stabilizes CL at the inner mitochondrial membrane and inhibits cytochrome c mediated CL peroxidation, protected from DKD in vivo [9]. In another study using db/db mice, it was demonstrated that accumulation of total renal lysocardiolipin is associated with DKD, while the use of Elamipretide had renoprotective effects [115]. Interestingly, a role of innate immune complement component C5a in the cardiolipin remodeling and reduction of mitochondrial fatty acids metabolism in DKD has been demonstrated using mouse models of type 1 (Ins2-Akita mice, streptozotocin-induced diabetic mice and rats) and type 2 diabetes (db/db mice) [116]. While CL seems to be a significant player in DKD development and progression, it remains unclear if CL contributes to mtDNA escape into cytosol in renal cells.

### 3.3. Mitophagy

Mitophagy is a selective form of autophagy, where damaged or dysfunctional mitochondria undergo degradation and recycling. The PTEN-induced putative kinase protein 1 (PINK1)/E3 ubiquitin–protein ligase (Parkin) pathway is the most studied mechanism of mitophagy. Loss of mitochondrial membrane potential results in PINK1 accumulation at the outer mitochondrial membrane (OMM), phosphorylation (on Ser65) of pre-existing ubiquitin molecules and Parkin recruitment. In turn, PINK1-dependent phosphorylation of the ubiquitin-like domain of Parkin leads to the release of catalytic RING2 domain, which stabilizes Parkin in its functionally active state, followed by ubiquitination of other OMM proteins including voltage-dependent anion-selective channel (VDAC), mitochondrial Rho GTPase proteins (MIRO), mitofusin 1 (MFN1) and 2 (MFN2) [117]. Parkin promotes ubiquitination of LC3 on the lysine 63 (K63) and lysine 48 (K48) residues, whereby K48 ubiquitination initiates passive mitochondrial degradation, while K63 ubiquitination leads to the recruitment of the autophagy adaptors LC3/GABARAP (Figure 2). The underlying mechanisms, however, are not fully understood and remain to be further elucidated.

Other mechanisms of mitophagy have been also described. For example, the FUN14 domain containing 1 (FUNDC1) protein has a conserved LC3-interacting region. Under hypoxia or loss of mitochondrial membrane potential, dephosphorylation of Tyr18 and Ser13, mediated by the mitochondrial phosphatase PGAM family member 5 (PGAM5), and concomitant phosphorylation of Ser17 by ULK1 enhances FUNDC1 and LC3 interaction to promote mitophagy. Activity of PGAM5 is thereby controlled by BCL2-like 1 (better known as BCL-XL) [118]. FUNDC1 was found to interact with both the mitochondrial fission key factor dynamin 1 like (DRP1) and the inner membrane fusion regulator OPA1 to coordinate mitochondrial dynamics and mitophagy [119].

BCL2 interacting protein 3 (BNIP3) and NIX are proteins localized at the outer mitochondrial membrane and are also involved in stress sensing and hypoxia-induced mitophagy [120]. An increase in BNIP3 protein levels leads to the liberation of Beclin 1 from BCL2 apoptosis regulator and BCL-XL sequestration to initiate mitophagy and prevent mtROS production and cell death. Both BNIP3 and NIX interact with LC3 to further enhance autophagosomal recruitment to mitochondria [121].

Impaired mitophagy is recognized as a hallmark of human DKD and of rodent models of DKD. In patients with DKD and in rats with streptozotocin-induced DKD, activation of thioredoxin interacting protein (TXNIP) under hyperglycemic conditions was shown to cause accumulation of autophagosomes and reduced autophagic clearance in tubular cells [122]. Renal tubular epithelial cells treated with high glucose and biopsies from DKD patients have shown reduced levels of mitophagy [123]. Diabetic db/db mice were found to have decreased expression levels of mitochondrial PINK1, Parkin, LC3-II, Beclin-1 and Atg5, all the markers of impaired mitophagy [124]. Progression of DKD is associated with gradual decrease of Parkin expression in renal tubular epithelial cells of patients with DKD (n = 149), and overexpression of Parkin reduces inflammation and improves renal function in streptozotocin-induced diabetic mice [125]. Similar, reduced PINK1/Parkin mitophagy was reported under high glucose conditions in HK-2 cells and in the streptozotocin-induced mouse model of DKD [10,126,127], in proximal tubular cells [122] and in podocytes and db/db mouse model of DKD [128] and in rats on a high fat diet [129]. In contrast, some studies report an abnormal activation of PINK1/Parkin mediated mitophagy in db/db mice [130,131]. As no differences in the genetic background of the animals used in these studies have been noticed, more detailed investigations are needed to explain the opposite results observed in the activation pattern of the PINK1/Parkin mediated autophagy.

Interestingly, involvement of the PI3K/AKT/mTOR signaling pathway, a major intracellular network that regulates cell proliferation and life cycle and in the regulation of autophagy in glomerular mesangial cells have been shown in several studies [129,132,133,134,135,136], suggesting an important role of autophagy in renoprotection. As in other kidney cell types, decreased levels of LC3-II, PINK1 and Parkin have been reported in glomerular endothelial cells under high glucose treatment [128]. The use of different compounds to regulate mitophagy in DKD has been widely reported and is summarized in Table 1.

### 3.4. Pyroptosis and Ferroptosis

Pyroptosis is a caspase-1-dependent form of cell death that is triggered by proinflammatory signals from microbial infections and non-infectious stimuli. While caspase-1-dependent cell death is mediated by caspases, it was initially not distinguished from apoptosis. However, it has become clear that the mechanism, characteristics and outcome of caspase-1-dependent cell death are distinct from apoptosis [140]. Pyroptosis can also be initiated by non-canonical inflammasome pathway via activation of caspase-11 in mice and caspase-4 in humans and the cleavage of gasdermin D (GSDMD) [141]. Notably, TLRs, RLRs and NLRs have been shown to mediate pyroptosis, whereby NLRP3 is the most connected molecule to pyroptosis as reviewed elsewhere [142,143]. Caspase-4/11 and GSDMD-dependent pyroptosis contribute to podocyte loss and DKD progression [144]. In high glucose treated podocytes and streptozotocin-induced DKD mice, NLRP3-mediated upregulation of GSDMD and mtROS/NLRP3 dependent pyroptosis have been described [145]. Interestingly, in membranous nephropathy, another type of glomerular disease, complement-induced pyroptosis was shown to contribute to podocytes injury [146]. Intriguingly, cleavage of the amino-terminal sequence of GSDMD induces mitochondrial outer membrane permeabilization [147,148,149], but the mechanism needs to be further elucidated.

Ferroptosis is a novel form of programmed cell death derived by the iron-dependent peroxidation of lipids through the cysteine/glutamate antiporter Xc- (xCT) and glutathione peroxidase 4 (GPX4)-dependent mechanisms. In the state of high fructose, significant upregulation of mitochondrial single-strand DNA-binding protein 1 (SSBP1) has been shown to contribute to podocyte injury via activation of the transcriptional factor p53 and ferroptosis [150]. In the mouse glomerular podocyte MPC5 cell line, high glucose was found to induce ferroptosis via suppression of peroxiredoxin 6 (Prdx6), an antioxidant that reduces oxidative stress, and specificity-protein 1 (Sp1), zinc finger family transcription factor, regulating cell survival and proliferation in many ways [151]. Interestingly, a significant role of VDAC, the mitochondrial transmembrane channel that transports ions and metabolites, plays an important regulatory role in ferroptosis via ROS- and nitric oxide-dependent signaling pathways [152]. Moreover, lipid metabolism, which is dysregulated in DKD, is closely associated with ferroptosis, and phosphatidylethanolamine is the key phospholipid that induces ferroptosis in cells. Lipid induced ROS accumulation is another mechanism leading to ferroptosis, and mitochondria have been shown to contribute to lipid induced ROS accumulation in mouse embryonic fibroblasts [153], suggesting a crucial role of mitochondria in ferroptosis. A correlation between iron, lipid peroxidation and ferroptosis associated marker acyl-CoA synthetase long-chain family member 4 (ACSL4) was established in renal tubular cells of db/db and streptozotocin-induced DKD mice [154]. Ferroptosis may also contribute to DKD development via suppression of nuclear factor-erythroid factor 2-related factor 2 (NRF2) [155], a critical transcriptional factor involved in the regulation of many cellular processes. Transforming growth factor β (TGFβ)-stimulated tubular cells also exhibit increased levels of ferroptosis, which were shown to be reduced by ferrostatin-1, the ferroptosis inhibitor [156].

Thus, there remains no doubt that pyroptosis and ferroptosis contribute to podocyte death in mouse models of DKD, but the exact mechanistic pathways have not yet been identified. Moreover, whether ferroptosis contributes to the progression of DKD in patients with diabetes remains to determined.

## 4. Conclusions

Although mitochondria are master regulators of inflammation and cell death in the diabetic kidney (Figure 2), additional research is needed to address many questions. Thus, well-established mechanistic links between inflammatory responses directed by mitochondria and DKD development and progression are often missing and remain to be uncovered. Moreover, additional work is needed to characterize the roles of autophagy and apoptotic caspases in the regulation of inflammation driven by mitochondrial damage associated molecular pattern, with special attention to the roles of other cellular processes associated with mitophagy in different kidney cells in DKD. Additionally, the key molecular details, such as the interaction of mtDNA with inflammasomes or of mtDNA depletion and altered cellular biogenesis and oxidative balance remain to be clarified. Surprisingly, little progress has been made over the past 10 years in uncovering the specific roles of the important transcriptional factor NF-κB in mitochondrial function and data revealing the presence of NF-κB in mitochondria from renal cells are missing. Therefore, the discovery of a role for NF-κB signaling in mitochondria in DKD may open new therapeutic perspectives. Additionally, the levels of ROS have not been carefully measured in podocytes, and applying a single-cell RNA sequence approach should be used to eliminate this issue. Lastly, it remains necessary to continue the investigation of mitochondrial function under physiological and pathological conditions which will ultimately lead to the discovery of novel therapeutics to prevent, reverse and treat DKD and, possibly, other diabetic complications.

## Figures and Tables

**Figure 1 cells-11-03635-f001:**
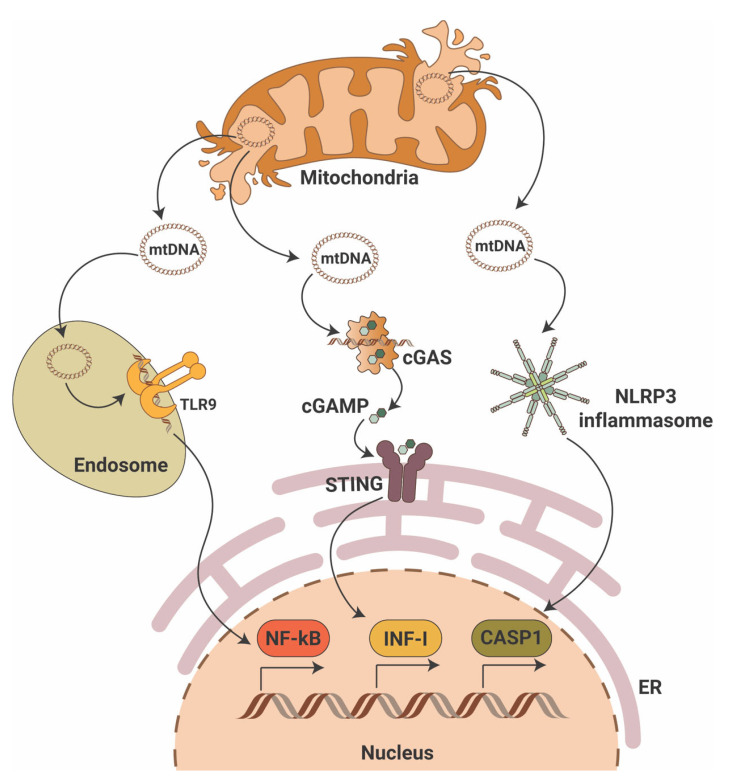
Mitochondrial DNA triggers pro-inflammatory signaling pathways. Mitochondrial DNA (mtDNA) can activate three major pro-inflammatory signaling pathways that include endosomal toll-like receptor 9 (TLR9) with activation of nuclear factor κB (NF-κB), the cyclic GMP-AMP synthase (cGAS)/stimulator of interferon gene (STING) with activation of interferon type I (INF-I) and cytosolic NLR family pyrin domain containing 3 (NLRP3) inflammasome activation with induction of caspase-1-dependent apoptosis. ER—endoplasmic reticulum; cGAMP—2′3′-cyclic GMP-AMP.

**Figure 2 cells-11-03635-f002:**
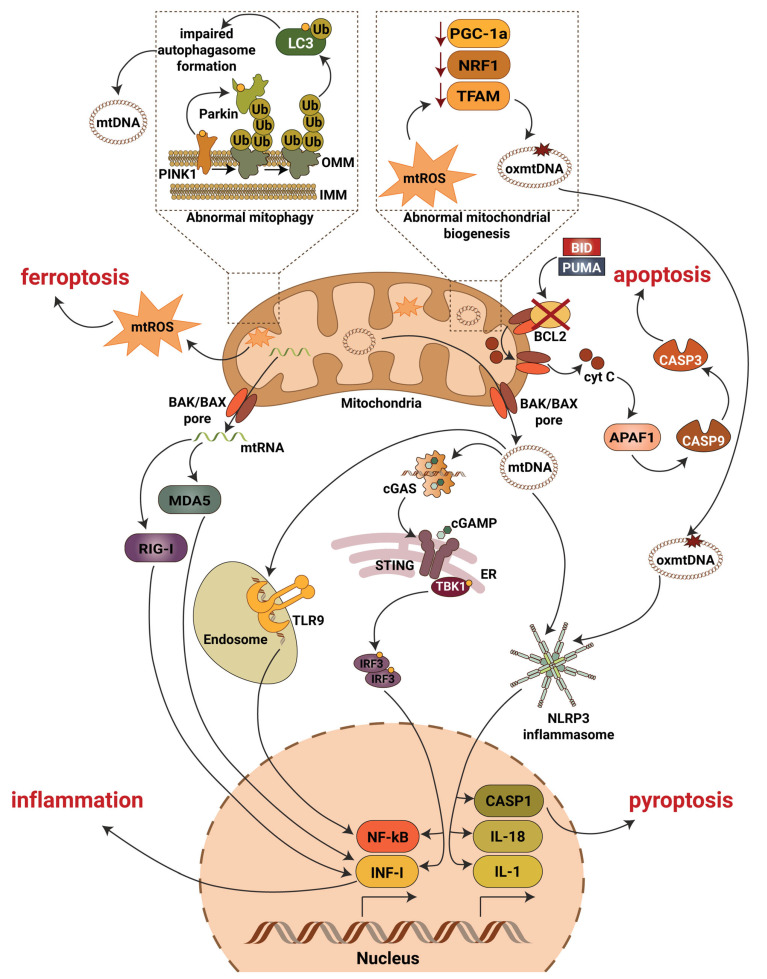
Mitochondrial contribution to DKD associated kidney damage. Generation of mitochondrial reactive oxygen species (mtROS), escape of mitochondrial RNA (mtRNA) and mitochondrial DNA (mtDNA) into cytosol, production of oxidized mitochondrial DNA (oxmtDNA) and abnormalities in mitophagy lead to activation of major pro-inflammatory signaling pathways, pyroptotic, apoptotic and ferroptotic cell death. Abbreviations: APAF1—apoptotic pepdidase activating factor 1; BAK—Bcl2 homologous antagonist/killer; BAX—Bcl2-associated X protein; BCL2—B-cell lymphoma 2; BID—BH3 interacting-domain death agonist; CASP1—caspase 1; CASP3—caspase 3; CASP9—caspase 9; cGAS—cyclic GMP-AMP synthase; cGAMP—2′3′-cyclic GMP-AMP; Cyt C—cytochrome c; ER—endoplasmic reticulum; IL-1—interleukin 1; IL-18—interleukin 18; IMM—inner mitochondrial membrane; INF-I—interferon type I; IRF3—interferon regulatory factor 3; LC3—microtubule-associated protein 1A/1B-light chain 3; MDA5—melanoma differentiation-associated protein 5; NF-kB—activation of nuclear factor κB; NLRP3—NLR family pyrin domain containing 3; NRF1—nuclear respiratory factor 1; OMM—outer mitochondrial membrane; PGC-1a—peroxisome proliferator-activated receptor-γ coactivator-1α1; PINK1—PTEN induced kinase 1; PUMA—p53 upregulated modulator of apoptosis; RIG-I—retinoic acid-inducible gene I; STING—stimulator of interferon gene; TBK1—TANK binding kinase 1; TFAM—mitochondrial transcription factor A; TLR9—toll-like receptor 9; Ub—ubiquitin.

**Table 1 cells-11-03635-t001:** Mitophagy modulators in experimental models of DKD.

Compound	Target	Model	Phenotype	Ref.
Metformin		HFD mice	↓ mtROS production	
	AMPK	STZ mice	↑ PINK1 and Parkin protein levels	[137]
		HK-2 cells	↑ LC3-II and Atg5 levels	
Mitotempo	mtROS	HG-treated	↓ mtROS production	[123]
		mouse primary RTECs	↓ p16, p21, SAHF, SA-β-Gal, DcR2	
MitoQ			↓ oxidative stress	
	Nrf2/PINK1	db/db mice	↓ caspase-3 expression	[138]
			↑ Δψm	
D-glucarate		HK-2 cells	Restored mitochondrial morphology	
	MIOX	STZ mice	↓ mtROS production	[10]
			↓ BAX mediated apoptosis	
CoQ10		HG-treated		[128]
	Nrf2	mGECs	↑ mitophagy via PINK1 and Parkin	
Triptolide			↑ autophagy and LC3-II levels	[129]
	PTEN/AKT/mTOR	HFD rats	↓ p62	
Palmitic acid			↑ mitophagy and LC3 levels	
	PINK1/Parkin	HFD rats	↑ mtROS production	[139]
			↑ apoptosis	
Progranulin		HFD rats		[127]
	CAMKK-AMPK	STZ mice	↑ autophagy and LC3 levels	
APF		HG-treated RMCs		[136]
	mTOR/PINK1/Parkin	STZ mice	↑ mitophagy via PINK1 and Parkin	
Icariin		STZ rats	↓ superoxide anion	[133]
	Nrf2/GPER	HG-treated hGMCs	↑ antioxidant enzymes activity	
Ursolic acid			↑ LC3 levels	[135]
	PI3K/AKT/mTOR	HG-treated rGMCs	↑ PTEN mRNA and protein expression	
Astragaloside IV			↓ mitophagy via PINK1 and Parkin	[130]
	Not specified	db/db mice	↓ renal DRP1, FIS1, MFF expression	
HDD	Not specified	db/db mice	↓ mitophagy via PINK1 and Parkin	[131]

Abbreviations: AKT—protein kinase B; AMPK—AMP-activated protein kinase; APF—Astragalus mongholicus Bunge and Panax notogiseng F.H. Chen formula; CAMKK—calcium/calmodulin-dependent protein kinase; DRP1—dynamin-related protein 1; FIS1—mitochondrial fission 1 protein; hGMCs—human glomerular mesangial cells; rGMCs—rat glomerular mesangial cells; mGEC—murine glomerular endothelial cells; GPER—G protein-coupled estrogen receptor; HDD—Huangqi-Danshen decoction; HFD—high fat diet; HG—high glucose; HK-2—human kidney 2 cells; MFF—mitochondrial fission factor; MIOX—Myo-inositol oxygenase; mTOR—mammalian target of rapamycin; PT—proximal tubular cells; PTEN—phosphatase and tensin homolog; RMCs—renal mesangial cells; RTECs—renal tubular epithelial cells; STZ—streptozotocin; Δψm—mitochondrial membrane voltage potential.

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
