# Peer review of "Mitochondrial Contribution to Inflammation in Diabetic Kidney Disease"

_cells, 2022, doi:10.3390/cells11223635_

Round 1

Reviewer 1 Report

The review by Mitrofanova et al nicely addresses the expansive topic of multiple inflammatory signaling pathways related to mitochondrial dysfunction in the context of diabetic kidney disease. The criticisms below should mainly be viewed as suggestions (rather than demands) to improve the review quality.  

Major criticisms

11.      The implication throughout the review is that ROS are derived from mitochondria. This is controversial (PMID 27217197), in part because ROS have not always been carefully measured in podocytes. Now that there are multiple scRNAseq databases, including for diabetic kidney disease, the authors should address this issue, and provide more detail to support the veracity of ROS origin from mitochondria.   

22.      There is extensive discussion about the gain of mtDNA in the cytosol, which triggers disease-causing signaling pathways. Please provide any evidence of an alternative hypothesis that kidney cell dysfunction is due to loss of mtDNA, resulting in concomitant mitochondrial dysfunction.  

33.      There should be some discussion about diminished mitochondrial biogenesis and fusion, particularly in context of the inflammation theme.

Minor criticisms

11.      Replace reference 4 with a more contemporary citation that reflects residual renal disease in the SGLT2 inhibitor and/or finerenone era.

22.      Lines 96-97, please clarify the meaning of mtDNA lies in close proximity to mtROS. The unintended implication is that there is a physical interaction.   

33.      Delete n=12 from line 126 and n=11 from line 224.

44.      Data from multiple cell types are mixed throughout the review. To draw clearer conclusions it would be helpful to distinguish/group citations for podocytes, other glomerular cells, tubule cells, interstitial cells, whole kidney, cortex, etc. Alternatively the authors might want to more clearly state situations where cell specificity is/is not important.

55.      Line 216, “form” should be “from”.  

66.      In lines 262-268, might want to be absolutely clear that VDAC is no longer considered to be a component of the mitochondrial transition pore complex, since this has been controversial until recently. Rather, emphasize that a novel role for VDAC may be as a pore for mtDNA.

Author Response

Dear Reviewer,

Reviewer 2 Report

In this review, Authors discussed the role of mitochondria in the activation of inflammatory responses and cell death initiation specifically in podocytes under disease conditions (i.e. diabetic kidney disease) and suggested some potential areas for future investigation. As a general comment, the manuscript is well organized, fluent and covers the relevant literature. In the first part, Authors mainly focused on the contribution of released mtDNA as a trigger factor for inflammation and kidney injury and explored all downstream signaling pathways. As recognized by the Authors, the mechanisms of mtDNA release that would imply an increased permeation of both inner and outer membranes still remain elusive. In this regard, Authors discussed the involvement of Bax/Bak pores and suggested a possible implication of cardiolipin in this process. I believe that Authors should include in the list also the Permeability Transition Pore (PTP), the mitochondrial megachannel (only briefly mentioned in lines 264-267) that has been proposed to play a role in mtDNA release and be therefore a critical key factor in the process. Although its involvement in the pathological outcomes of DKD is controversial (Lindblom RSJ et al., 2020 Clin Sci. doi: 10.1042/CS20190787), it was shown to be more active in kidney mitochondria from diabetic rats (Oliveira PJ et al., 2004 Diabetes Metab Res Rev. doi: 10.1002/dmrr.423). More recently, data revealed that its inhibition by TRAP1, the mitochondrial chaperone, ameliorates diabetes-induced renal injury (Liu L et al., 2020 Oxid Med Cell Longev. doi: 10.1155/2020/6431517). Therefore, I would suggest discussing this aspect in more detail. 

Author Response

Dear Reviewer,
